# Primary Total Prostate Cryoablation for Localized High-Risk Prostate Cancer: 10-Year Outcomes and Nomograms

**DOI:** 10.3390/cancers15153873

**Published:** 2023-07-30

**Authors:** Chung-Hsin Chen, Chung-You Tsai, Yeong-Shiau Pu

**Affiliations:** 1Department of Urology, National Taiwan University Hospital, Taipei 10002, Taiwan; chunghsinchen2@ntu.edu.tw; 2Division of Urology, Department of Surgery, Far Eastern Memorial Hospital, New Taipei City 22000, Taiwan; pgtsai@mail.femh.org.tw; 3Department of Electrical Engineering, Yuan Ze University, Taoyuan City 32003, Taiwan

**Keywords:** cryotherapy, nomogram, outcome prediction, biochemical failure, prostate malignancy, recurrence

## Abstract

**Simple Summary:**

The role of prostate cryoablation was still uncertain for patients with high-risk prostate cancer (PC). This study was designed to investigate the 10-year outcomes and establish a nomogram for high-risk PC patients. We found prostate cryoablation to be an effective treatment option for selected men with high-risk PC. A preoperative nomogram that predicts biochemical recurrence would be useful for both patients and physicians to make clinical decisions when considering prostate cryoablation among other treatment modalities. A peri-operative nomogram that includes diagnostic PSA, PSA nadir, Gleason sum, and the number of cryoprobes deployed helps inform increased risk of biochemical recurrence, which would then justify early salvage treatments.

**Abstract:**

The role of prostate cryoablation was still uncertain for patients with high-risk prostate cancer (PC). This study was designed to investigate 10-year disease-free survival and establish a nomogram in localized high-risk PC patients. Between October 2008 and December 2020, 191 patients with high-risk PC who received primary total prostate cryoablation (PTPC) were enrolled. The primary endpoint was biochemical recurrence (BCR), defined using Phoenix criteria. The performance of pre-operative and peri-operative nomograms was determined using the Harrell concordance index (C-index). Among the cohort, the median age and PSA levels at diagnosis were 71 years and 12.3 ng/mL, respectively. Gleason sum 8–10, stage ≥ T3a, and PSA > 20 ng/mL were noted in 27.2%, 74.4%, and 26.2% of patients, respectively. During the median follow-up duration of 120.4 months, BCR-free rates at 1, 3, 5, and 10 years were 92.6%, 76.6%, 66.7%, and 50.8%, respectively. The metastasis-free, cancer-specific, and overall survival rates were 89.5%, 97.4%, and 90.5% at 10 years, respectively. The variables in the pre-operative nomogram for BCR contained PSA at diagnosis, clinical stage, and Gleason score (C-index: 0.73, 95% CI, 0.67–0.79). The variables in the peri-operative nomogram for BCR included PSA at diagnosis, Gleason score, number of cryoprobes used, and PSA nadir (C-index: 0.83, 95% CI, 0.78–0.88). In conclusion, total prostate cryoablation appears to be an effective treatment option for selected men with high-risk PC. A pre-operative nomogram can help select patients suitable for cryoablation. A peri-operative nomogram signifies the importance of the ample use of cryoprobes and helps identify patients who may need early salvage treatment.

## 1. Introduction

Localized prostate cancer (PC) can be managed via several treatment options, including radical prostatectomy (RP), radiation therapy (RT), cryoablation, high-intensity focused ultrasound, and active surveillance/watchful waiting [1]. Among them, cryoablation is less recommended for patients with localized high-risk PC defined by at least one component of prostate-specific antigen (PSA) > 20 ng/mL, Gleason grade group of 4 or 5, and clinical T stage of T2c or more [1,2]. However, its advantages of short hospital stay, minimal anesthesia, and rapid recovery due to the minimally invasive nature provide benefit to aged patients or those with multiple comorbidities [2,3,4,5]. Furthermore, focal cryoablation in highly select patients leads to few adverse events and preserves most functional outcomes [6,7]. In the aspect of oncological outcomes, a satisfactory 5-year biochemical recurrence (BCR)-free rate of 62.2% in patients with high-risk PC was reported by the Cryo On-Line Database (COLD) Registry, the largest database regarding prostate cryoablation in the world [3]. Although the treatment failure rate of primary total prostate cryoablation (PTPC) was significantly higher in patients with high-risk compared to intermediate- or low-risk PC [3], it was also high in high-risk PC treated with RP (5-year disease-free survival (DFS) rate, 38–65%) [8,9] or RT plus androgen deprivation therapy (ADT) (5-year DFS rate, 62–74%) [8,9]. PTPC may be still feasible for selected patients with high-risk PC. Until recently, long-term (10-year) oncological outcomes for high-risk PC patients and nomograms predicting recurrences were still lacking. To provide better clinical decision-making, we herein report the cohort of PTPC for patients with high-risk PC.

## 2. Materials and Methods

### 2.1. Patient Population

Between October 2008 and December 2020, consecutive patients with localized PC who received PTPC at National Taiwan University Hospital were prospectively collected. Bone scintigraphy and multi-parametric magnetic resonance imaging (MRI) were applied for the initial tumor staging in all patients. Among them, only patients with high-risk disease defined by EAU guidelines 2023 [2] were enrolled in the current study. (Flow diagram in Appendix A) This study was reviewed and approved by Research Ethics Committee A of National Taiwan University Hospital (202204097RINA). We previously published the short-term results of our entire patient cohort, including non-high-risk disease [4].

### 2.2. Clinical Information Collection

Clinicopathological data regarding patient age, prostate size measured via transrectal ultrasound, pre-operative PSA, biopsy Gleason sum, clinical T stage, tumor location in MRI, neoadjuvant ADT, the amount of cryoprobes used intraoperatively, follow-up PSA values, time to BCR, recurrence patterns, and survival were prospectively collected. Clinical T stage was determined by means of either digital rectal examination or seminal vesicle biopsy prior to PTPC. Nine patients with clinical T3b disease were defined according to the result of seminal vesicle biopsy. Neoadjuvant ADT that usually took 4–12 weeks was mainly to reduce prostate size when the anterior–posterior diameter exceeded 35 mm to facilitate the cryoablation procedure. Twenty patients received adjuvant ADT under a clinical trial setting [10].

All cryoablation procedures were performed by the single surgical team, Drs. CH Chen and YS Pu. The detailed surgical procedures were described in the previous published article [11]. All patients were followed using the same protocol, including PSA every 3 months in the first year, every 6 months in the 2nd to 5th years, and then annually. The primary outcome was BCR, which was determined using the Phoenix criteria, i.e., PSA increase of ≥2 ng/mL above the nadir [12]. Upon BCR, we commended early restaging and early salvage therapy for these high-risk PC patients based on the consensus of the European Association of Urology Prostate Cancer Guidelines Panel [13]. Hence, we advised prostate and seminal biopsy, whole-body computed tomography/MRI imaging, and/or whole-body PET-CT scan (^18^F-choline or ^68^Ga-PSMA) to distinguish between local and distant failure. For patients with negative biopsy or unwilling to have a prostate biopsy, a whole-body PET-CT scan (^18^F-choline or ^68^Ga-PSMA) was conducted as an alternative. The primary outcome was BCR determined using the Phoenix criteria calculated from the serial follow-up PSA value.

### 2.3. Statistical Consideration

Contingency tables were constructed for comparisons using the Chi-square test. Nonparametric data were compared with the Mann–Whitney U rank-sum method to compare medians between groups. The log-rank test and Cox proportional hazards model were used to compare the BCR risk. All these analyses were conducted using R software, version 3.6.1 (http://www.r-project.org/, accessed on 1 September 2021). All tests were two-tailed with *p* < 0.05 indicating a significant difference.

The original patient cohort was randomly split into two cohorts: one (80% of patients) served as the training cohort for developing the predictive prognostic models, and the other (20%) as the validation cohort for external validation (Appendix A). Univariable and multivariable Cox proportional hazard models were applied to address the time to BCR after cryoablation. Multivariable Cox regression coefficients were used to generate prognostic nomograms. We intended to generate two prognostic nomograms: pre- and peri-operative predictive nomograms. Only pre-operative variables were incorporated into the pre-operative nomogram, while both pre- and peri-operative variables were used in the peri-operative nomogram. The final models were selected using a bidirectional stepwise regression process, which used the Akaike information criterion as a stopping rule [14]. The nomograms were constructed using the *survival* and *rms* packages in R [15].

The model performance for predicting the BCR-free survival was determined using the Harrell concordance index (C-index) [16]. To avoid the arbitrariness of cohort splitting, the bias-corrected C-index was further calculated using repeated five-fold cross-validation 20 times and 1000 bootstrapping methods for the entire cohort. Calibration plots were constructed to compare the nomogram-predicted probability of BCR-free survival at 1, 3, 5, and 7 years with the actual survival probability.

## 3. Results

### 3.1. Patient Demographics and Tumor Characteristics

A total of 233 consecutive PC patients who received PTPC were enrolled. Forty-two subjects were excluded because of low- or intermediate-risk disease. The median age of the remaining 191 high-risk PC patients was 71 years (range 48–88 years), and the median PSA value at diagnosis was 12.3 ng/mL (range 2–45.9 ng/mL; Table 1). The median time to BCR was 34.5 months and the 5-year BCR-free rate was 66.7%. There was PSA < 10, 10–20, and >20 ng/mL for 40.3%, 33.5%, and 26.2% of patients, respectively. About half of the patients had a Gleason sum ≥ 4 + 3 (49.7%). Clinical T3a or above occurred in 74.4% of patients. Visible tumor lesions on MRI (PI-RADS 3–5) were noted in 83.3% of patients. Tumors located at the anterior apical prostate, which might be difficult to be treated [17], were identified in 29 (15.2%) patients. Neoadjuvant ADT for ≤3 and >3 months was applied in 94 (49.2%) and 9 (4.7%) patients, respectively. Twenty (10.5%) patients were given adjuvant ADT for 12 months under a prospective randomized study. Figure 1 shows that the higher the number of high-risk factors based on EAU guidelines 2023 (PSA, Gleason, and stage) that patients had, the faster the BCR occurred.

### 3.2. Univariable and Multivariable Analyses Predicting BCR

Among the 191 patients, 111 (58.1%) remained BCR-free after a median follow-up duration of 120.4 months (IQR 63–137.7 months). Compared to patients without BCR, those with BCR (*n* = 80) tended to be younger (median 69 vs. 72 years, *p* = 0.014) and have higher PSA at diagnosis (median 15.4 vs. 10.0 ng/mL, *p* < 0.001), higher Gleason sum (8–10, 37.5% vs. 19.8%, *p* = 0.004), and higher clinical T stage (≥T3b 46.3% vs. 22.5%, *p* = 0.006) (Table 1). There was no significant difference in prostate size, amount of cryoprobes used, the proportion of visible lesions on MRI, anterior apical tumors, or proportion of subjects receiving neoadjuvant ADT and adjuvant ADT between patients with and without BCR.

After PTPC, 77.0% of patients reached the PSA nadir within 3 months post-operatively. Eighty-seven (45.6%) patients had a PSA nadir value < 0.01 ng/mL. The PSA nadir values were significantly higher in men with subsequent BCR than those without (*p* < 0.001). For patients with a PSA nadir ≥ 0.5 ng/mL, up to 10 (91%) out 11 patients experienced BCR. In comparison, only 24.1% (21/87) of patients had BCR if the post-cryoablation PSA nadir was <0.01 ng/mL.

In the multivariable analysis of the pre-operative parameters, higher PSA, higher Gleason sum, and higher clinical T stage independently predicted BCR (Table 2). For the peri-operative predictive model, the multivariable analysis revealed that significant independent predictors for BCR included higher PSA, higher Gleason sum, fewer or inadequate number of cryoprobes used, and higher PSA nadir value (Table 2). More cryoprobes used appeared to lower the risk of BCR, suggesting that the effective coverage of cancer areas through adequately overlapping the cryoablation kill zone was crucial for treating high-risk PC. The PSA nadir value was the most powerful predictor in the peri-operative predictive model for BCR. Compared to patients with PSA nadir < 0.01 ng/mL, those with a nadir of 0.01 to <0.1 (hazard ratio (HR) = 2.98, 95% confidence interval (CI) 1.68–5.26), 0.1 to <0.5 (HR = 6.32, 95% CI 3.26–12.3), and ≥0.5 ng/mL (HR = 37.98, 95% CI 15.5–93.1) had significantly elevated risk of BCR. Although the time to PSA nadir was associated with BCR in the univariable analysis, it was a non-significant factor for BCR in the multivariable model because of the strong association with PSA nadir (Appendix A).

### 3.3. Predictive Nomograms and Calibration

To predict BCR-free survival probability, pre- and peri-operative nomograms (Figure 2) were constructed according to the multivariable predictive models. Three parameters were included in the pre-operative nomogram: PSA at diagnosis, biopsy Gleason sum, and clinical T stage. Four parameters were incorporated in the peri-operative nomogram: PSA at diagnosis, PSA nadir, biopsy Gleason sum, and number of cryoprobes used.

In the training cohort, the C-indexes for the pre- and peri-operative nomograms were 0.74 (95% CI 0.67–0.79) and 0.82 (95% CI 0.76–0.88), respectively. For the validation cohort, C-indexes were 0.76 (95% CI 0.61–0.91) and 0.84 (95% CI 0.69–0.99), respectively. Bias-corrected C-indexes for the two nomograms were 0.70 and 0.80, respectively. The calibration plots showed satisfactory agreement in the BCR-free survival probabilities calculated from either the nomograms or the actual survival data for both pre- and peri-operative nomograms (Figure 3).

### 3.4. Pathological and Radiographic Evidence of Recurrence

The BCR-free rate at 1, 2, 3, 5, 7, and 10 years was 92.6%, 84.5%, 76.6%, 66.7%, 59.5%, and 50.8%, respectively (Figure 1). Among the 80 patients with BCR, 44 (55%) local recurrences were detected via either prostate biopsy (*n* = 38) or imaging studies (*n* = 6). Metastasis to the pelvic lymph node, bone, and both were found in 13 (16.3%), six (7.5%), and one (1.3%) patient, respectively. The remaining 16 (20.0%) patients who had BCR did not have any pathological or radiographic evidence of local recurrence or distant metastases. The estimated 10-year metastasis-free rate was 89.5% using Kaplan–Meier method. There was no visceral metastasis upon the identification of BCR. Two patients had lung metastases in 58 and 64 months after BCR. Three of the 191 patients had died of PC by the date of report preparation (January 2023). Eleven patients died of cardiovascular diseases, infection and other cancers. The estimated 10-year cancer-specific and overall survival rates were 97.4% and 90.5%.

### 3.5. Complications

A total of 44 (23.0%) patients had complications after PTPC in this cohort. The most common complication was bladder outlet obstruction (*n* = 30, 15.7%), which included bladder neck contracture (*n* = 8, 4.2%), urethral stricture (*n* = 8, 4.2%), urethral sloughing (*n* = 10, 5.2%), urethral stone (*n* = 1, 0.5%), and mixed type (*n* = 3, 1.6%). Eighteen (9.4%) patients had infection-related events, such as epididymitis (*n* = 3, 1.6%), prostatitis (*n* = 8, 4.2%), and urethrocystitis (*n* = 7, 3.7%). Among 81 patients with potency before PTPC, 9 (11.1%) recovered their erectile function with or without medication. No cryoprobe penetration wound infection was noted. Long-term urinary incontinence was observed in five (2.6%) patients. Transfusion was used for two (1.0%) patients. One patient encountered a suspected sigmoid injury which was handled with parenteral nutrition for 7 days and was discharged without any sequelae.

## 4. Discussion

Total prostate cryoablation is an alternative treatment option for localized PC, especially for low- and intermediate-risk disease [5,18]. Our data showed, in terms of BCR after a long-term follow-up duration, that PTPC provided adequate 10-year cancer control in 50.8% of patients with high-risk disease, which is generally comparable with historical control using other treatment modalities, such as RP [8,9] or RT [8,9]. The conventional pre-operative clinicopathological parameters, including PSA at diagnosis, Gleason sum, and clinical T stage helped predict BCR after PTPC. In the peri-operative setting, except for PSA at diagnosis and Gleason sum, PSA nadir and number of cryoprobes used comprised a powerful predictive model for BCR. The two models or nomograms provide valuable tools to inform clinical decision-making and prognostic information in PTPC. In addition, the peri-operative nomogram may help not only identify men at an increased risk of failure but also advise early salvage therapy.

Compared to lower risks, patients with high-risk disease have increased local recurrence and treatment failure rates regardless of treatment modality applied [1,8,9]. Therefore, the European Association of Urology guidelines suggested physicians offer multimodal therapy for the patients with high-risk localized PC [19]. Retrospective studies reported that the 5-year DFS rate with RP and RT was 38–65% and 62–74%, respectively [8,9]. In comparison, the COLD Registry and our series of PTPC demonstrated comparable outcomes of 62% and 66.7% for the 5-year BCR-free survival rate, respectively [3]. Although the definition of treatment failure differs between treatment modalities, that for RT and PTPC use the same Phoenix criteria. The BCR-free rates for high-risk disease between PTPC in our cohort and RT plus ADT [9] were similar at 5 (66.7% vs. 72%) and 10 years (46% vs. 53%), respectively. Since differences in demographics and tumor characteristics between patient populations may significantly affect clinical outcomes, it is inappropriate to compare these numbers directly. For example, clinical T3 disease was more frequent in our cohort (74%) than the RT cohort (14%). In contrast, the RT cohort had a higher proportion of PSA ≥ 20 ng/mL (36% vs. 26%) and Gleason score 8–10 (41% vs. 27%) compared with our PTPC cohort.

The addition of long-term adjuvant ADT to definitive RT has become a standard-of-care option for localized high-risk PC [20,21]. It was not clear whether adjuvant ADT would benefit patients receiving PTPC. In a small-scale (*n* = 38) prospective randomized study, adjuvant ADT for 12 months did not reduce BCR in patients with high-risk PC receiving PTPC [10]. The benefit of adjuvant ADT in PTPC may be minimal or uncertain and should be further investigated in large-scale studies. In our series, 103 (53.9%) had received neoadjuvant ADT to reduce prostate size before cryoablation. Only nine patients received neoadjuvant ADT for more than 3 months. However, our univariable analysis showed that neoadjuvant ADT did not significantly influence BCR-free survival (*p* = 0.126). In addition, we found no significant association between neoadjuvant ADT and PSA nadir after cryoablation (Appendix A).

The PSA nadir values have been identified as an important prognostic factor for clinical outcomes after PTPC. Tay et al. reported that patients with PSA nadir < 0.4 ng/mL in the COLD Registry had a significantly better 5-year DFS of about 70%, compared to those with PSA nadir ≥ 0.4 ng/mL where most patients failed within 5 years [22]. The multivariable analysis revealed that PSA nadir was a powerful prognosticator for BCR [22]. Many other Western reports [22,23] and our data on Asian men also showed consistent results in that PSA nadir values significantly predicted the treatment failures after PTPC. We also found that the earlier the PSA nadir was reached (<8 weeks), the higher the chance of BCR, suggesting that post-cryoablation residual cancer nests drove PSA recurrence and shortened the time to PSA nadir.

The number of cryoprobes used was a significant predictor of BCR in the multivariable peri-operative regression model. In general, the number of cryoprobes to be deployed in PTPC depends on several factors, including but not limited to prostate size, shape, and specific tumor locations [17]. Saturated prostate cryoablation via setting more cryoprobes and reducing the prostate volume to be covered per probe would improve cancer control [4]. A higher number of cryoprobes will reduce any possible inadequate ablation zone in the prostate and ensure that the overlapping ice balls in the prostate reach a substantially low killing temperature. These findings suggest that proactive and ample use of cryoprobes to cover as complete a prostate region as possible may reduce inadequate ablation zone and subsequent recurrence, especially when dealing with high-risk tumors.

Total prostate cryoablation had an acceptable rate of side effects in our patients with high-risk PC. Although infection-associated complications, such as epididymitis, prostatitis, and urethrocystitis, were up to 9.4%, all patients recovered well using the appropriate antibiotics. As a minimally invasive surgery [24], no infection at the penetration wound of the cryoprobes and thermoprobes was noted in our patients. The long-term continence rate, defined as 1 or less pad a day, was 97.4% and comparable to those of RP and RT series [25,26]. Bladder outlet obstruction resulting from bladder neck contracture, urethral stricture, and urethral sloughing was relatively higher in our high-risk PC patients than low- to intermediate-risk PC patients [3]. The possible reason was the intention to ablate as much of the prostate as possible and, subsequently, to break the protection zone of the urethral warming catheter in high-risk PC patients. Nevertheless, all these patients experienced improvement using endourological methods. In our high-risk cohort, we did not observe the most serious complication, namely, rectourethral fistula. A possible sigmoid injury was noted during prostate cryoablation in one patient, who was supported with total parenteral nutrition for one week and discharged without any sequelae. Considering high-risk PC patients, the complication rate of PTPC was acceptable and comparable to a non-nerve-sparing prostatectomy or radiation therapy plus androgen deprivation, which were considered as the preferred treatment options [1]. We did not identify significant clinical predictors for complications after PTPC in our series. The major reason was the patient selection bias. For example, we did not conduct PTPC in patients whose tumors were located near the urethra or who ever had transurethral resection of the prostate. To evaluate the possible predictors of complications from PTPC, a prospective cohort without significant selection criteria is warranted.

Several limitations exist in our study. First, the case number was not large enough for an extensive analysis of all clinical parameters in the multivariable model. However, the important demographic and tumor phenotype variables were all incorporated into the final model for establishing the predictive nomograms. Second, there was a lack of an independent cohort for external validation of the models or nomograms—this was well compensated using the bootstrap and cross-validation in our study. Third, this is a retrospective analysis that may have selection bias. However, the study enrolled all consecutive patients who received total prostate cryoablation, and all data variables were prospectively collected for all patients in a well-designed data entry file from the start of the study, which may significantly mitigate any selection bias or recall bias. Fourth, the current nomogram included only clinical parameters, but not molecular and detailed histological characters. Although this design made it convenient for the physician to use in clinical practice, the precision of outcome prediction may increase with more molecular/histological biomarkers, such as serine/arginine splicing factor 1 [27], microvessel density [27], insulin growth factor-1 [28], and so on.

## 5. Conclusions

Total prostate cryoablation appears to be an effective treatment option for men with high-risk PC. A pre-operative nomogram that predicts BCR would be useful for both patients and physicians to make clinical decisions when considering cryoablation among other treatment modalities. A peri-operative nomogram that includes diagnostic PSA, PSA nadir, Gleason sum, and the number of cryoprobes deployed may help inform increased risk of BCR, which would then justify early salvage treatments.

## Figures and Tables

**Figure 1 cancers-15-03873-f001:**
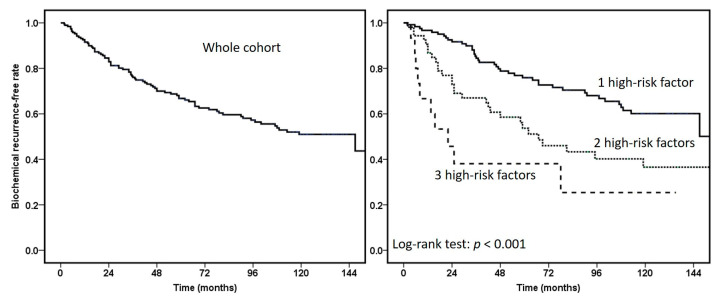
The Kaplan–Meier curve of biochemical failures in the high-risk prostate cancer patients receiving primary total prostate cryoablation. The high-risk factors were defined based on EAU guidelines 2023 and included Gleason sum of 8 or more, PSA value of 20 ng/mL or more, and clinical stage T3a or more.

**Figure 2 cancers-15-03873-f002:**
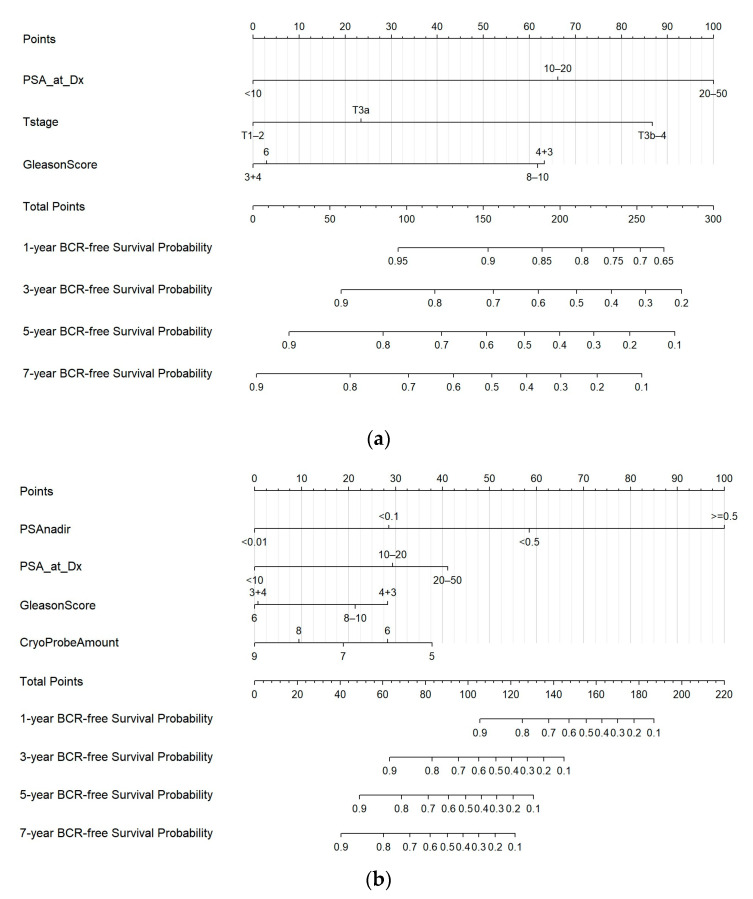
The nomograms for predicting biochemical recurrence in the high-risk or very high-risk prostate cancer patients: pre-operatively (**a**) and peri-operatively (**b**).

**Figure 3 cancers-15-03873-f003:**
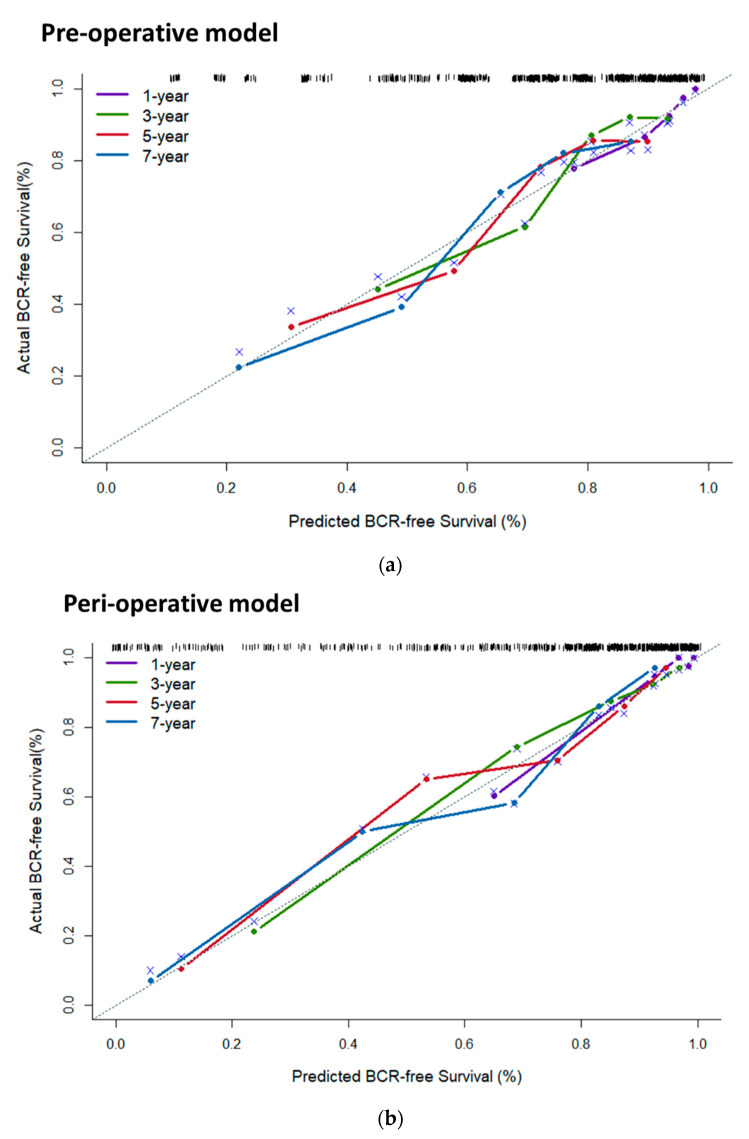
The calibration of the nomograms predicting biochemical failures: pre-operatively (**a**) and peri-operatively (**b**). BCR = biochemical recurrence. Remark “x”: resampling optimism added.

**Table 1 cancers-15-03873-t001:** Demographics of high-risk prostate cancer patients stratified by the status of biochemical failure.

Groups	All		No BCR		BCR		*p* Value
Patient number (n)	191	100.00%	111	58.1%	80	41.90%	
Median age (years, range)	71 (48–88)		72 (52–88)		69 (48–87)		0.014
Median PSA at diagnosis (ng/mL, range)	12.3 (2.0–45.9)		10.0 (2.0–45.9)		15.4 (4.5–44)		<0.001
PSA at diagnosis (ng/mL)							0.003
<10	77	40.31%	56	50.45%	21	26.25%	
10~20	64	33.51%	33	29.73%	31	38.75%	
>20	50	26.18%	22	19.82%	28	35.00%	
Biopsy Gleason sum							0.004
≤6	35	18.32%	26	23.42%	9	11.25%	
3 + 4 = 7	61	31.94%	42	37.84%	19	23.75%	
4 + 3 = 7	43	22.51%	21	18.92%	22	27.50%	
8~10	52	27.23%	22	19.82%	30	37.50%	
Clinical T stage							0.006
T1c	8	4.19%	5	4.50%	3	3.75%	
T2a-2c	41	21.47%	29	26.13%	12	15.00%	
T3a	80	41.88%	52	46.85%	28	35.00%	
T3b	62	32.46%	25	22.52%	37	46.25%	
Visible lesions on MRI							0.084
No	32	16.75%	23	20.72%	9	11.25%	
Yes	159	83.25%	88	79.28%	71	88.75%	
Anterior apical tumor							0.449
No	162	84.82%	96	86.49%	66	82.50%	
Yes	29	15.18%	15	13.51%	14	17.50%	
Neoadjuvant hormonal therapy							0.153
No	88	46.07%	56	50.45%	32	40.00%	
Yes	103	53.93%	55	49.55%	48	60.00%	
Adjuvant hormonal therapy							0.437
No	171	89.53%	101	90.99%	70	87.50%	
Yes	20	10.47%	10	9.01%	10	12.50%	
Prostate volume (median in mL, range)	26.9 (11.9–81.9)		26.9 (11.9–81.9)		26.8 (12.6–64.0)		0.689
Cryoprobe number (median, range)	6 (5–9)		6 (5–8)		6 (5–9)		0.351
PSA nadir value (ng/mL)							<0.001
<0.01	87	45.55%	66	59.46%	21	26.25%	
0.01~<0.1	66	34.55%	35	31.53%	31	38.75%	
0.1~<0.5	27	14.14%	9	8.11%	18	22.50%	
0.5~	11	5.76%	1	0.90%	10	12.50%	
Time to PSA nadir (weeks)							0.084
<8	71	37.17%	34	28.57%	37	51.39%	
8~<12	76	39.79%	48	40.34%	28	38.89%	
12~	44	23.04%	29	24.37%	15	20.83%	

BCR = biochemical recurrence; PSA = prostate-specific antigen; CI = confidence interval; MRI = magnetic resonance imaging.

**Table 2 cancers-15-03873-t002:** Univariable and multivariable analyses of biochemical recurrence in high-risk prostate cancer patients receiving primary total prostate cryoablation.

			Univariable Analysis			Multivariable Analysis					
	Case No.	Failure Events				Preoperative Model			Peri-Operative Model		
Variables			HR	Range	*p* Value	HR	Range	*p* Value	HR	Range	*p* Value
Pre-operative											
Age (year)	191	80	0.98	0.95–1.01	0.145	▬	▬	▬	▬	▬	▬
PSA at diagnosis (ng/mL)								0.001 *			<0.001 *
<10	77	21	1	▬	▬	1	▬	▬	1	▬	▬
10~<20	64	31	2.01	1.15–3.50	0.014	1.93	1.10–3.38	0.022	2.48	1.38–4.46	0.002
20~<50	50	28	3.15	1.78–5.57	<0.001	2.75	1.54–4.91	0.001	3.68	1.91–7.09	<0.001
Biopsy Gleason sum								0.001 *			0.001
~6	35	9	1	▬	▬	1	▬	▬	1	▬	▬
3 + 4 = 7	61	19	1.1	0.50–2.43	0.817	1.11	0.50–2.46	0.804	1.03	0.45–2.34	0.941
4 + 3 = 7	43	22	2.26	1.04–4.92	0.039	2.3	1.05–5.02	0.037	2.90	1.31–6.37	0.009
8~10	52	30	3.01	1.49–6.64	0.003	2.4	1.12–5.13	0.024	2.44	1.13–5.24	0.023
Clinical T stage								0.007			
T1c-2c	49	15	1	▬	▬	1	▬	▬	▬	▬	▬
T3a	80	28	1.07	0.57–2.00	0.833	1.08	0.57–2.04	0.809	▬	▬	▬
T3b	62	37	2.62	1.43–4.78	0.002	2.18	1.18–4.00	0.012	▬	▬	▬
Visible lesions on MRI											
Yes	159	71	1	▬	▬	▬	▬	▬	▬	▬	▬
No	32	9	0.52	0.56–1.04	0.063	▬	▬	▬	▬	▬	▬
Anterior apical tumor											
No	162	66	1	▬	▬	▬	▬	▬	▬	▬	▬
Yes	29	14	1.31	0.74–2.33	0.361	▬	▬	▬	▬	▬	▬
Neoadjuvant hormonal therapy											
No	88	32	1	▬	▬	▬	▬	▬	▬	▬	▬
Yes	103	48	1.42	0.91–2.22	0.126	▬	▬	▬	▬	▬	▬
Prostate volume (mL)	191	80	1	0.98–1.02	0.908	▬	▬	▬	▬	▬	▬
Post-operative											
Cryoprobe number	191	80	0.87	0.70–1.10	0.239				0.69	0.53–0.92	0.002
Adjuvant hormonal therapy											
No	171	70	1	▬	▬	▬	▬	▬	▬	▬	▬
Yes	20	10	1.1	0.57–2.14	0.778	▬	▬	▬	▬	▬	▬
PSA nadir value (ng/mL)											<0.001 *
<0.01	87	21	1	▬	▬				1	▬	▬
0.01~<0.1	66	31	2.37	1.36–4.14	0.002				2.98	1.68–5.26	<0.001
0.1~<0.5	27	18	6.56	3.46–12.5	<0.001				6.32	3.26–12.3	<0.001
0.5~	11	10	28.44	12.7–63.6	<0.001				37.98	15.5–93.1	<0.001
Time to PSA nadir (weeks)											
<8	71	37	1	▬	▬				▬	▬	▬
8~<12	77	28	0.52	0.32–0.85	0.009				▬	▬	▬
12~	43	15	0.48	0.27–0.89	0.019				▬	▬	▬

PSA = prostate-specific antigen; HR = hazard ratio; MRI = magnetic resonance imaging; * *p* value of the variables which were considered to be ordered.

## Data Availability

Data are available on request due to restrictions of privacy consideration of our institutions. The data presented in this study are available on request from the corresponding author. The data are not publicly available due to the consideration of patients’ privacy.

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
