# Peer review of "Primary Total Prostate Cryoablation for Localized High-Risk Prostate Cancer: 10-Year Outcomes and Nomograms"

_cancers, 2023, doi:10.3390/cancers15153873_

Round 1
Reviewer 1 Report (Previous Reviewer 2)
The manuscript has been improved very well.
Reviewer 2 Report (Previous Reviewer 1)
The manuscript has been sufficiently improved according to reviewer's suggestions. It is now suitable for publication in Cancers.
This manuscript is a resubmission of an earlier submission. The following is a list of the peer review reports and author responses from that submission.
Round 1
Reviewer 1 Report
The authors demonstrated that Prostate Cryo was an effective treatment possibility for patients with high-risk PC, selected through preoperative nomogram. The manuscript sounds well; the experiments are well performed and the results well presented. I have some minor concerns:
- SRSF-1 protein and the histologic count of microvessel density were correlated with poorer prognosis and increased cell proliferation (doi: 10.1002/pros.24185)
The authors should discuss these findings.
Reviewer 2 Report
In this study, authors investigated 10 years-outcomes after cryoablation in patients affected by localized high-risk prostate cancer. This study is really interesting and relevant even because FT is an emerging therapeutic option for localized prostate cancer, although its role is still uncertain in high-risk diseases.
The manuscript is well-written and fluent but it needs some revisions of the language before being suitable for publication.
Please redefine D’Amico high-risk group with the latest EAU guidelines definition of only high-risk
Please define locoregional, did you mean localized?
Please correct the main indication in high-risk PCa in the Introduction section (doi: 10.3389/fsurg.2023.1157528)
Cryoablation is a minimally-invasive treatment that is proposed also to reduce the note side effects of conventional therapies (doi: 10.3390/curroncol29100538). Please improve this aspect in the introduction section
In the result you describe clinical T3b, how did you diagnose them? Only with digital rectal exploration or thanks to MRI? Please report as clinical T only that with DRE otherwise specify it
Authors should describe how BCR was managed after treatment (doi: 10.1016/j.euf.2019.06.004)
Erectile function wasn’t analyzed as a functional outcome after treatment and should be assessed in the results section
Authors could report in the discussion section if there is a correlation between complications found after treatment and the histopathological location (at the biopsy) of the cancer and/or with the number of cryoprobes used
The abbreviations section should be reported
English shall be revised by a native speaker